# Clinical Significance of Tumor Infiltrating T-Helper and Regulatory Cells in Bulgarian Cervical Cancer Patients

**DOI:** 10.3390/biomedicines13092206

**Published:** 2025-09-09

**Authors:** Angel Yordanov, Polina Damyanova, Mariela Vasileva-Slaveva, Konstantina Karakadieva, Stoyan Kostov, Velizar Shivarov

**Affiliations:** 1Department of Gynecologic Oncology, Medical University-Pleven, 5800 Pleven, Bulgaria; angel.jordanov@gmail.com (A.Y.); kkarakadieva@gmail.com (K.K.); 2Department of General and Clinical Pathology, Heart and Brain Center of Clinical Excellence, 5800 Pleven, Bulgaria; poldamdim@abv.bg; 3Research Institute, Medical University-Pleven, 5800 Pleven, Bulgaria; sscvasileva@gmail.com (M.V.-S.); drstoqn.kostov@gmail.com (S.K.); 4Department of Breast Surgery, Shterev Hospital, 1000 Sofia, Bulgaria; 5Department of Gynecology, Hospital “Saint Anna”, Medical University—“Prof. Dr. Paraskev Stoyanov”, 9002 Varna, Bulgaria

**Keywords:** CD4+ T-helper cells, FOXP3+ regulatory T cells, cervical cancer, tumor progression

## Abstract

**Background and Objectives:** Cervical cancer (CC), primarily caused by human papillomavirus (HPV) infection, is the most common gynecological cancer and a leading cause of cancer-related death in women. The immune microenvironment, particularly CD4+ T-helper cells and FOXP3 (forkhead box P3)+ regulatory T cells (Tregs), plays a crucial role in tumor progression. However, the exact relationship between immune cell infiltration and clinical outcomes in CC is not fully understood. This study aimed to examine the association between CD4+ T-helper cells, FOXP3+ Tregs, and clinical/pathological parameters in CC patients. Methods: We conducted a retrospective analysis of 150 patients with T1-stage cervical cancers diagnosed between 2015 and 2021. Tumor samples were evaluated using immunohistochemistry (IHC) to assess CD4+ and FOXP3+ TILs in intratumoral and stromal regions. Additionally, deconvoluted transcriptomic data from the TCGA cohort were used to assess immune infiltration within the tumor immune microenvironment (TIME). **Results:** High infiltration of CD4+ T-helper cells was significantly associated with better overall survival (OS) in node-negative CC patients (*p* = 0.0006). However, no significant prognostic value was found for Tregs. CD4+ cells were more prevalent in patients with well-differentiated tumors (G1 and G2) and lower levels of CD4+ infiltration were found in squamous cell carcinoma (SCC) compared to other histological subtypes. Multivariate regression analysis showed that only tumor size (T1b3) and undifferentiated tumor morphology (G3) were significantly associated with poorer OS. In contrast, infiltration of CD4+ or FOXP3+ cells did not significantly correlate with OS after adjusting for clinical factors. Competing risk analysis for death from cancer showed no significant associations with immune cell infiltration levels. **Conclusions:** This study underscores the complex relationship between immune cell infiltration and clinical outcomes in CC. While CD4+ T-helper cell infiltration is associated with improved prognosis in node-negative cases, further research is necessary to clarify the role of Tregs and other immune components in the tumor microenvironment. These findings suggest potential avenues for therapeutic strategies, including immune checkpoint inhibitors, in CC.

## 1. Introduction

Cervical cancer (CC) is the most common gynecological cancer and ranks fourth in cancer morbidity and mortality among women [1,2]. According to the GLOBOCAN 2022 report, in 2022, CC was the eighth most frequently diagnosed malignancy worldwide with a total of 661,021 newly diagnosed cases and the ninth most lethal localization accounting for 348,189 deaths [3]. More than 70% of the mortality from CC is observed in developing countries, where it ranks second in morbidity and mortality from oncological diseases among women [4]. This is due to the lack of vaccination and screening programs in these regions [5,6].

About 99% of cases of cervical cancer are due to human papillomavirus (HPV) infection, with 70% associated with HPV types 16 (HPV-16) and 18 (HPV-18) [7]. Carcinogenesis in this disease is a long process, passing through HPV infection, progressing to precancerous lesions and invasive cancer, which can last up to 15 years [2]. Carcinogenesis is influenced by the dynamics and composition of the immune microenvironment, and this process is called “immunoediting”. Due to its viral genesis, CC has a higher lymphocytic infiltration compared to HPV-negative malignancies [2]. HPV-infected cancer cells can modulate the immune microenvironment and thus create a pro-tumorigenic state of immune suppression and evasion through multiple mechanisms. In one of these mechanisms, CD4+ T-helper cells play a major role, with a shift to a Th2 response being observed, which potentiates the humoral immune response. Another mechanism is the formation of HPV antigen-specific regulatory T (Treg) cells, whose purpose is to suppress both CD8+ and CD4+ cells in the tumor microenvironment [8]. Treg cells mainly express forkhead transcription factor (FOXP3) protein, which is associated with the activation of CD4+ CD25+ Treg cells in some tumors [9].

The aim of this study was to determine whether there is a relationship between intratumoral and stromal infiltration by CD4+ and FOXP3+ Tregs and some pathological parameters in patients with CC.

## 2. Materials and Methods

### 2.1. Patients and Tissue Samples

This is a retrospective study that included paraffin-embedded tumor tissues from 150 patients with CC diagnosed between 2015 and 2021 at the Pathology Clinic of Medical University-Pleven (Pleven, Bulgaria). All cases were T1 sized tumors either without (N0) or with (N1) local lymph nodes involvement (Table 1). We used materials available in the hospital’s pathology archive that contained a sufficient amount of tumor tissue and where the study would not lead to their depletion or damage. Permission was received from the Ethics Commission (number 656/29 June 2021). Clinico-morphological data for the patients were obtained from the electronic database of the Department of Gynecologic Oncology, where all patients underwent primary surgery. We collected data on patient age, the tumor stage at diagnosis according to the International Federation of Gynecology and Obstetrics (FIGO) and TNM classifications (The 8th edition of the TNM classification and FIGO classification 2018) [10,11], the tumor size, lymphovascular space invasion (LVSI) and the dedifferentiation stage (Table 1). In FIGO stage III, only patients with lymph node metastases are included (FIGO IIIC)—according to the current guidelines for treatment of CC, primary surgery is not recommended for patients with FIGO stage IIIA and FIGO stage IIIB.

### 2.2. Pathological Examination

One histological slide with hematoxylin and eosin staining, and a corresponding paraffin block were selected for each patient to assess the tumor-infiltrating lymphocyte (TIL) population. We prepared slides for staining with immunohistochemistry (IHC) to evaluate the CD4 and FOXP3.

For the IHC study, a visualization system, EnVision™ FLEX, High pH (DAKO), AutostainerLink 48 technique (DAKO), and the following primary antibodies were used: CD4 (clone 4B12, Mo, RTU, Dako, Glostrup, Denmark) and FOXP3 (clone 236A/E7, Mo, 1:100, Abcam, Cambridge, UK). All work procedures were met according to the respective antibody protocols of the manufacturer. The scoring of positive IHC stained cells for CD4 and FOXP3 was performed independently by two pathologists who had no information about the clinical and pathological data of the studied cases. Cells with membrane expression for CD4 and nuclear expression for FOXP3 were counted as positive. They detected and reported positivity only in lymphocytes, as tumor cells were negative for the applied antibodies and were therefore not included in our analysis. The average value of the results of the studied patients with cervical cancer was taken as the cut-off.

Initially, the entire tumor area in each IHC stained slide was examined. Then, CD4+ and FOXP3-positive lymphocytes were scored in the tumor and stroma of five randomly selected fields, at high magnification (high power fields—HPF), ×400, without focusing on a hot-spot when there were positive cells. An average number of up to 6 CD4 IHC-positive and up to 5 FOXP3-positive cells were reported as their low intratumoral concentration. Intratumoral CD4 and FOXP3 lymphocytes with scores equal to or greater than 6 and 5, respectively, were considered as high TILs intratumoral values. We also report a percentage of stromal areas occupied by CD4+ TILs (CD4+ TILs %) and an average number of stromal FOXP3+ lymphocytes. As a cut-off value, we accepted 10% for CD4 and a count of 15 for FOXP3-positive cells. Lower values (CD4 < 10% and FOXP3 < 15 in count) were accepted as low concentration, and equal or higher values (CD4 ≥ 10% and FOXP3 ≥ 15 in count) as high concentration of the specified lymphocyte subtypes. For the purpose of statistical processing of the obtained immunohistochemical results, the absence of intratumoral (IT) and stromal (ST) CD4 and FOXP3 lymphocytes, their low and high concentration were designated as 0, 1 and 2, respectively (Table 2). Examples of the IHC specified categories are illustrated in Figure 1, Figure 2, Figure 3 and Figure 4. Each picture has a stromal and tumor component, but only one of the two is indicated with an arrow, the one that is demonstrative of the corresponding concentration category of the evaluated type of lymphocyte. We also present the relevant photos with hematoxylin and eosin staining of the selected cases to better distinguish the separately assessed tumor and stromal components.

### 2.3. Deconvoluted Transcriptomic Data

In this study, transcriptomic data for cervical cancer were obtained from the consensus tumor immune microenvironment (TIME) dataset within human tumor immune micro-environment cell composition database (TIMEDB) [12], specifically derived from The Cancer Genome Atlas (TCGA) project [13]. Gene expression profiles from bulk RNA sequencing data were analyzed using conventional deconvolution methods to estimate the relative abundance of immune and stromal cell populations within the tumor microenvironment [12]. The consensus TIME dataset integrates multiple ten deconvolution algorithms to enhance the robustness of inferred cell-type compositions [12]. Data extraction was performed via the TIMEDB web interface (https://timedb.deepomics.org/, last accessed on 10 January 2025), ensuring standardized and reproducible access to processed transcriptomic datasets. The retrieved data were subsequently integrated with clinical and pathological parameters from TCGA to investigate associations between TME composition and patient outcomes as described below.

### 2.4. Statistical Analysis

All IHC markers were systematically cross-tabulated against key clinical and pathological parameters, including patient age (categorized with a threshold of 50 years), tumor (T) stage, nodal (N) stage, and histological subtype. The resulting contingency tables were analyzed using a Chi-squared test to account for the limited sample sizes within certain cells of the tables. A *p*-value < 0.05 was considered statistically significant.

To further assess associations, multivariate linear regression analyses were conducted for all TILs subpopulations, incorporating all available clinical and pathological parameters as covariates. Additionally, survival outcomes for patient subgroups stratified by IHC marker expression were evaluated using the Kaplan–Meier method. Statistical significance of differences between survival curves was assessed via the log-rank test. Kaplan–Meier survival curves were generated and analyzed using the *survminer* package (version 0.5.0) in R (version 4.5.0).

To determine the impact of clinical and pathological variables on overall survival, a multivariate Cox proportional hazards model was constructed, incorporating all clinical and pathological covariates. This analysis was performed using the survival package in R. Furthermore, cumulative incidence functions were estimated to differentiate the risk of mortality due to neoplastic disease from other causes, employing the Kaplan–Meier method. These functions were visualized using the *survminer* package (version 0.5.0).

For a more refined assessment of competing risks, a multivariate Fine–Gray model was applied to estimate the effect of covariates on disease-specific mortality, utilizing the riskRegression package (version 2025.05.20) in R. Results from multivariate statistical models were illustrated using either the *forestmodel* (version 0.6.2) or *forestplot* (version 3.1.7) packages in R to enhance interpretability.

All statistical analyses were conducted using R version 4.5.0 for Windows, ensuring reproducibility and robustness of the findings.

## 3. Results

### 3.1. Consensus Deconvolution of Transcriptomic Data to Demonstrate the Role of CD4+ TILs and Tregs in CC

We obtained consensus data for infiltration by CD4+ T cells and Tregs of CC samples included in the TCGA cohort through from the TIMEDB portal. The dataset also included clinical meta-data such as age, stage and histology (Figure 5). Notably, the level of infiltration by CD4+ T cells did not differ between squamous cell carcinoma (SCC) and non-SCC, but the infiltration by Tregs was significantly lower in SCC cases (*p* = 0.0007) (Figure 5A). On the other hand, high infiltration by CD4+ T cells was associated with longer overall survival (OS) (*p* = 0.0006) (Figure 5B). Higher infiltration by Tregs showed tendency to a better prognosis but failed to reach statistical significance (*p* = 0.057) (Figure 5C). In a multivariate Cox model, however, infiltration by neither CD4+ T cells nor by Tregs was associated with outcome (Figure 5D). In that model, only advanced stage disease was independently associated with shorter OS (*p* = 0.01) (Figure 5D).

### 3.2. Resolving Spatial Infiltration by CD4+ and FOXP3+ TILs by IHC

The analysis of deconvoluted transcriptomic data suggested that CD4+ and Tregs may play significant role in the clinical course of CC, but it could not be resolved fully because of the inherent limitations of the technique such as analysis of expression based on RNA and not on protein level and also lack of spatial resolution. Therefore, we aimed to overcome those limitations and to assess the infiltration by CD4+ and Tregs in our own CC cohort using conventional immunohistochemistry. We used only two standard markers for that purpose surface expression of CD4 and nuclear expression of FOXP3 by TILs (Figure 1 and Figure 2). We added spatial dimension to our data by separately assessing the infiltration by CD4+/FOXP3+ TILs, which are in contact with neoplastic cells (designated as intratumoral (IT) subpopulations) and by CD4+/FOXP3+ TILs infiltrating tumor stroma without being in direct contact with a malignant population (designated as stromal (ST) subpopulations) (Figure 1 and Figure 2). All subpopulations were assessed in a semiquantitative manner by H-score as described in Section 2.

### 3.3. Association of CD4+ and FOXP3+ TILs with Clinical Characteristics

We further analyzed the association of the four defined subpopulations of CD4+ (CD4IT and CD4ST) and FOXP3+ Tregs (FOXP3IT and FOXP3ST) with clinical features and with each other (Table 2). Analysis of association with the FIGO stage was obviously redundant as we included only operable T1 tumors which were either local lymph node positive or negative upon postoperative histological examination (see Section 2) (Table 1).

We did not observe any associations with tumor size (Table 2). However, we observed significantly lower infiltration by CD4IT (*p* < 0.001) and CD4ST (*p* < 0.00001) TILs in node-positive cases (Figure 6A,B). No such association was observed for FOXP3+ Tregs (Table 2). The presence of LVSI was also associated with slightly lower frequency of any level of infiltration by CD4ST (*p* = 0.014) (Figure 6C) (Table 2).

Interestingly, we identified some significant associations with levels of TILs and Tregs and features inherent to the malignant populations such as histological subtype and grade of differentiation. In that regard, we observed more CD4ST negative cases in patients with SCC than in adenocarcinoma (AC) and adenosquamous carcinoma (ASC) patients (*p* = 0.027) (Figure 7A). High-grade G3 tumors were also more frequently negative for CD4ST infiltration than G1 and G2 tumors (*p* = 0.032) (Figure 7B). On the other hand, it appeared that the highest proportion of cases negative for FOXP3ST infiltration was for SCC morphology (*p* = 0.00044) (Figure 7C). However, the highest proportion of cases negative for FOXP3IT infiltration was for AC morphology (*p* < 0.0001) (Figure 7D).

To further investigate the determinants of CD4+ TILs and Tregs infiltration, we used multivariate linear models with all clinical features and IHC parameters as covariates (Figure 8). CD4ST infiltration correlated linearly with CD4IT and FOXP3IT but did not correlate with any other parameter (Figure 8A). CD4IT correlated with ASC and SCC morphology and CD4ST and FOXP3IT but inversely with FOXP3IT infiltration (Figure 8B). FOXP3ST was inversely associated with SCC morphology but correlated linearly with CD4ST and FOXP3IT (Figure 8C). On the other hand, FOXP3IT infiltration correlated with SCC morphology and CD4IT and FOXP3ST (Figure 8D).

### 3.4. Associations Between CD4+ TILs and Tregs and Clinical Outcome

We further searched for any prognostic value of any of the infiltration of the four T cell subpopulations. In univariate analyses, only CD4IT infiltration had prognostic power with a clearly worse prognosis for patients without any infiltration (*p* = 0.01) (Figure 9A). High infiltration by CD4ST also probably confers a better prognosis but the overall log-rank test for CD4ST infiltration did not reach statistical significance (Figure 9B). Survival curves for all strata in infiltrations by FOXP3IT and FOXP3ST cell populations were overlapping without any tendency for statistical significance (Figure 9C,D). When we fitted the multivariate Cox regression model of OS with all available covariables, there was no significant parameter (Figure 10A). Only the largest tumor size (T1b3) and undifferentiated morphology (G3) were close to independent statistical significance with *p* = 0.07 and *p* = 0.06, respectively (Figure 10A). Indeed, tumor size and grade were significant prognostic factors for OS in univariate models (Figure 10B,C).

These results prompted us to investigate whether these indicators were associated with death from causes other than cancer. We performed a competing risk regression analysis, assuming death from cancer and other causes as competing risks. No difference was found between the cumulative hazard functions for the risk of death from neoplastic disease or other causes (Figure 11A). We analyzed the cumulative incidence of death from neoplastic disease using the Fine–Gray model (subdistribution hazard model) and found no statistically significant association between the individual indicators and the risk of death from neoplastic disease (Figure 11B).

## 4. Discussion

Over the last several decades, the role of immune mechanisms in cancer development has been established as core hallmarks of cancer, designated as tumor-promoting inflammation and evasion of immune destruction [14,15,16]. Evidently, immune cellular and humoral factors are a dynamic and co-evolving integral part of modulatory tumor environment [17], which is nowadays regarded as TIME [18]. Contemporary omics technologies provide the opportunity to investigate TIME in various settings in a systems immunology approach either through deconvolution of bulk expression data or single cell profiling approaches such as single cell RNA sequencing (scRNA-Seq), spatial transcriptomics, or mass flow cytometry [19]. However, classical low-resolution approaches based on immunohistochemistry are still the golden standard for implementation in clinical settings and validation of the findings from hypothesis-generating omics studies [20].

Recent large scale bulk RNA deconvolution studies suggested that squamous cell cervical cancer (CESC) is among the solid malignancies with the highest fraction of tumor infiltrating lymphocytes (TILs) [21]. Furthermore, the same study assigned most of the studies CESC cases from the TCGA project to the so called C2 cluster cancers [21]. The C2 subtype of cancers is considered IFN-gamma dominant and showed the highest M1 macrophages polarization, high infiltration by CD8+ T cells, high proliferation and diversity of the infiltrating T cells [21]. One may not consider surprising the predominance of CD8+ cytotoxic T cells in CESC as it is almost exclusively a virus-driven disease. However, the significance of other major subtypes of T cells, CD4+ is more elusive based on the abovementioned immune deconvolution approach. Of note, studies that tried to develop a CC-specific prognostic score using deconvoluted data did not include CD4+ T cell subpopulations in their immune score [22]. This might be due to the fact that the role of CD4+ TILs is context dependent and this context cannot be captured by conventional deconvolution approaches.

In support of the idea of a context-dependent role of TILs in cancer, a more sophisticated approach (called Ecotyper) for immune deconvolution and inference of tissue cellular communities (“ecotypes”) suggested a significant role of CD4+ T cells in SCC pathogenesis using the same TCGA dataset [23].

In order to translate this hypothesis to clinical settings, we explored the role of CD4+ TILs in limited stage CC patients from a single institution in Bulgaria using conventional immunohistochemical staining for CD4 and FOXP3. CD4 is the common marker used to define this major T cell population, and it is not expressed by malignant epithelial cells. On the other hand, FOXP3 is a transcription factor which defines a subset of CD4+ T cells with an immunosuppressive effector function known as regulatory T cells (Tregs). Interestingly, FOXP3 can be expressed in some CC and cervical intraepithelial neoplasia (CIN) cases [24,25,26]. However, we considered FOXP3 a suitable marker for IHC evaluation of Tregs infiltration in CC as TILs and cancer cells are easily distinguished by morphological features, as has already been shown for other solid cancers [27,28,29]. Most of the deconvolution approaches do not define T cell subsets. Therefore, we used the consensus deconvolution as described above to justify our selection of TILs subpopulations. Using data from the TCGA cohort, we observed correlations with histology of CC, a tendency for improved OS with higher content of CD4+ TILs and Tregs, which lost independent prognostic value in multivariate models. We considered these findings as justification for our IHC approach. In order to add some spatial resolution to our evaluation, we separately evaluated intratumoral and stromal infiltration by using semi-quantitative estimation in accordance with international recommendations for pathological assessment of TILs in solid cancers [30]. We have used this approach to quantify other immune cell subsets in cervical cancer [31,32,33].

Our evaluation of CD4+ TILs showed that their presence may correlate with the underlying biological features of the disease. In univariate models, infiltration with stromal and intratumoral CD4+ TILs appeared lower in node-positive disease, with a marginally higher absence of intratumoral CD4+ TILs in cases with LVSI. Notably, the lower infiltration with stromal CD4+ TILs was independently associated with nodal metastases as revealed by multivariate linear models. In the identical multivariate models, higher infiltration with intratumoral CD4+ TILs was independently associated with adenosquamous and squamous histology. In univariate overall survival analysis, higher intratumoral infiltration was associated with prolonged survival. However, its prognostic power was lost in multivariate models. To a large extent, our findings are in alignment with previous reports on CD4+ TILs in CC. An early study in limited stage SCC patients from Taiwan reported decreased infiltration with CD4+ TILs in patients with nodal metastases [34]. In that study, the reported infiltration by CD8+ TILs remained unaffected by the presence of nodal involvement and therefore it was suggested that low CD4/CD8 ratio of TILs might be the more accurate prognostic marker [34]. Piersma et al. showed that in HPV+ CC, lymph node-positive patients had significantly lower CD4/CD8 TILs ratio mainly because of a higher proportion of CD8+ cells [35]. This study and subsequent work suggested that the high CD8+/FOXP3+ Tregs ratio is a more important prognostic factor [35,36].

Several studies have shown that infiltration of cervical lesions by FOXP3+ Tregs increases gradually with the worsening of CIN status [37,38,39,40,41,42,43]. Furthermore, the level of infiltration with Tregs in SCC is even higher than in CIN I-III [41,43,44]. These data suggest that an increase in Tregs might be mechanistically involved in a progression from CIN to SCC due to enhanced immunosuppressive TME supported at least partly by the infiltrating Tregs [45]. In support of this idea are also the available reports of worse a prognosis in SCC in patients with higher FOXP3+ TILs [36,46]. In addition, other studies have shown that in cervical adenocarcinomas, the FOXP3+ Tregs may have beneficial effects and be associated with a better prognosis [47]. The prognostic value of infiltrating Tregs and CD4+ TILs might also be dependent on the volume of the primary tumor as some studies in more advanced CC showed improved survival with higher CD4+ and Tregs content [48]. None of these IHC-based studies, however, analyzed the spatial localization of infiltrating Tregs. Our work showed that this layer of pathobiological complexity is important, as in our multivariate models, stromal Tregs infiltration was significantly lower in SCC than in other histological subtypes. On the contrary, intratumoral Tregs infiltration was significantly higher in SCC. We did not observe other correlations of Tregs infiltration with any clinical features which might be due to the fact that the cohort was relatively uniform in terms of age and tumor size. In the multivariate Cox models, however, neither stromal nor intratumoral Tregs had independent prognostic value. In fact, to the best of our knowledge, only one histopathological study has used an approach identical to ours to analyze TILs in CC [49]. While Gultekin et al. did not analyze Tregs, they showed that only higher stromal CD4+ TILs infiltration was associated with better 5-year disease-free survival and with better distant metastasis-free survival [49]. The possible explanation for the localization specific effect of CD4+ TILs and Tregs is the complex quantitative relationships between different subpopulations as evidenced by our findings. Novel technologies such as scRNA-Seq and spatial transcriptomics have the potential to dissect the spatial relationships within TIME of CC in more depth. Several studies have already taken that approach [50,51,52,53,54,55,56,57,58,59,60,61]. Most of those studies were able to identify different T cell subpopulation based on conventional transcriptional programs. For example, Liu et al. processed eight SCC samples and divided CD4 into four CD4+ T subclusters: naïve CD4+ T cells (CCR7+); T-helper 17 (TH17) cells (IL17A+); TNFRSF9high regulatory T (Treg) cells (TNFRSF9high); and TNFRSF9low Treg cells (TNFRSF9low) [50]. Their results showed higher levels of TH17 and TNFRSF9high Treg infiltration in SCC compared to precancerous lesions and healthy controls, and that these levels increased with progression [50], which is consistent with previous IHC studies discussed above. Cao et al. divided CD4+ T cells into naive CD4+ T cells (CD4_CCR7), Th17 cells (CD4_IL17A), follicular helper T cells (Tfh, CD4_CXCL13), and regulatory T cells (Tregs, Treg_FOXP3) [58]. They concluded that TMC is dominated by Th17 and Tfh subsets, and also has a high density of FOXP3+ Tregs, suggesting an exhausted immune response [58]. Another study showed that naïve CD4 cells are more prevalent in cervical carcinoma than in precancerous low-grade squamous intraepithelial lesions (LSIL) and high-grade squamous intraepithelial lesions (HSIL), and that FOXP3-expressing Tregs are more abundant in cervical carcinoma but are in a state of exhaustion [60].

Very interesting results have been reported by authors analyzing the specific differences in the TME in SCC and AC. Peng et al. reported higher levels of Treg infiltration in AC compared to SCC, and also higher levels of Treg infiltration in HPV negative AC compared to HPV positive AC [51]. Lin et al. reported slightly different results—higher levels of Tregs infiltration in SCC compared to AC [59]. These results suggest that it is the HPV antigen-specific Tregs that may have immunosuppressive role in HPV-driven cancers. This is consistent with previous reports that HPV antigen-specific Tregs were associated with an inferior outcome [62]. Indeed, polyclonal HPV-antigen specific CD4+ and CD8+ are identified in both the primary tumor and draining lymph nodes in approximately half of patients and are poised for cytotoxic action upon proper modulation of TIME to help them overcome the exhausted phenotype [63]. This was supported by recent profiling of the TIL single cell repertoire, which showed that FOXP3+ Tregs from cancer tissues have increased clonality compared to Tregs from normal tissues, presumably as a response to HPV-specific antigens [58]. On the other hand, the true immunosuppressive function of Tregs in CC might be predominantly confined to a subset of activated highly immunosuppressive HLA-DR+ Tregs [64].

The population of CD4+ Th cells comprise heterogeneous subgroups and some authors reported different roles of these subgroups in cervical cancer (65–67). This is one of the limitations of this study—we detected only CD4 (not the different subgroups) and FOXP3 expression. We used IHC although there are more advanced technologies available, but IHC is more suitable for clinical use.

## 5. Conclusions

In sum, our own findings are in conjunction with previous studies which suggest that the role of CD4+ TILs and their subset of Tregs is context dependent and influenced by tumor-intrinsic features (e.g., histology, grade of differentiation, invasivness, HPV-status) and the dynamic composition of the TIME. Our study sheds some light on this complexity because of its obvious strengths, such as the large number of patients with small sized primary tumors, representing the three major histological variants of CC and the spatial resolution of TILs infiltration by analyzing both stromal and intratumoral lymphocytes. Unfortunately, we did not have available data on the HPV-status. Finally, this work suggested the feasibility of our approach and pointed to the necessity to extend the study to an integrated IHC analysis of other cellular components of the TIME in CC. This necessity is further supported by recent advances in the treatment of CC with immune checkpoint inhibitors [65,66], which most likely overcome the limitations of the immunosuppressive TIME in CC presumably through modulation of the activity of CD4+ TILs [45].

## Figures and Tables

**Figure 1 biomedicines-13-02206-f001:**
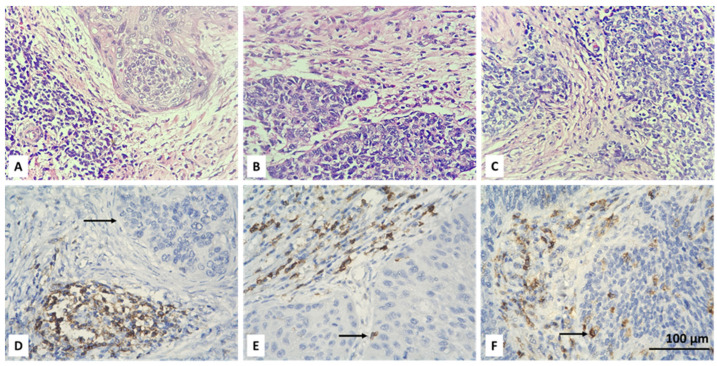
Microscopic evaluation of IHC-stained CD4+ T cells (colored brownish) in CC: absent (**D**), low (**E**) and high (**F**) intratumoral infiltration of positive cells (black arrow); corresponding hematoxylin and eosin stains are above (**A**–**C**); magnification ×400.

**Figure 2 biomedicines-13-02206-f002:**
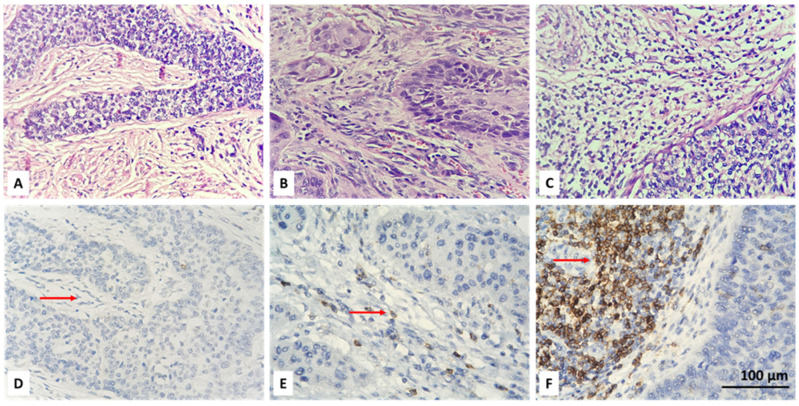
Microscopic evaluation of IHC-stained CD4+ T cells (colored brownish) in CC: absent (**D**), low (**E**) and high (**F**) stromal infiltration of positive cells (red arrow); corresponding hematoxylin and eosin stains are above (**A**–**C**); magnification ×400.

**Figure 3 biomedicines-13-02206-f003:**
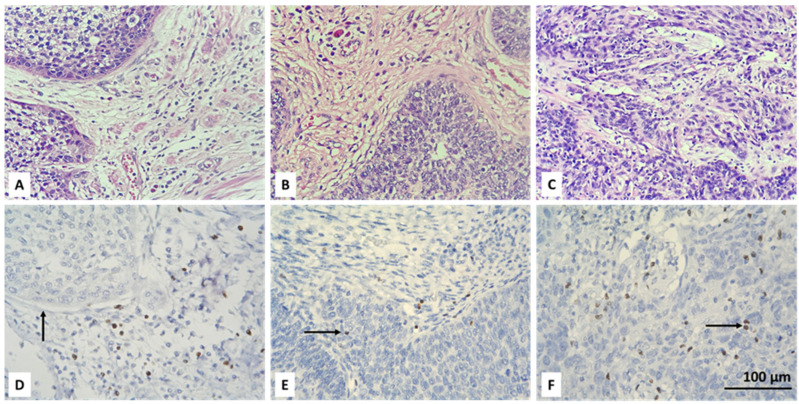
Microscopic evaluation of IHC-stained FOXP3+ T cells (colored brownish) in CC: absent (**D**), low (**E**) and high (**F**) intratumoral infiltration of positive cells (black arrow); corresponding hematoxylin and eosin stains are above (**A**–**C**); magnification ×400.

**Figure 4 biomedicines-13-02206-f004:**
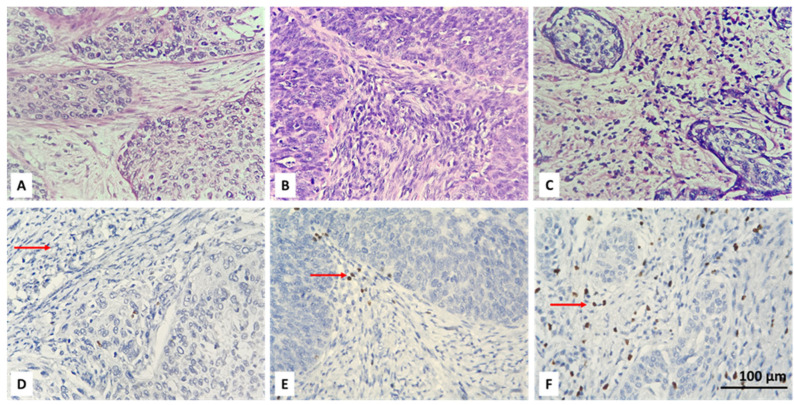
Microscopic evaluation of IHC-stained FOXP3+ T cells (colored brownish) in CC: absent (**D**), low (**E**) and high (**F**) stromal infiltration of positive cells (red arrow); corresponding hematoxylin and eosin stains are above (**A**–**C**); magnification ×400.

**Figure 5 biomedicines-13-02206-f005:**
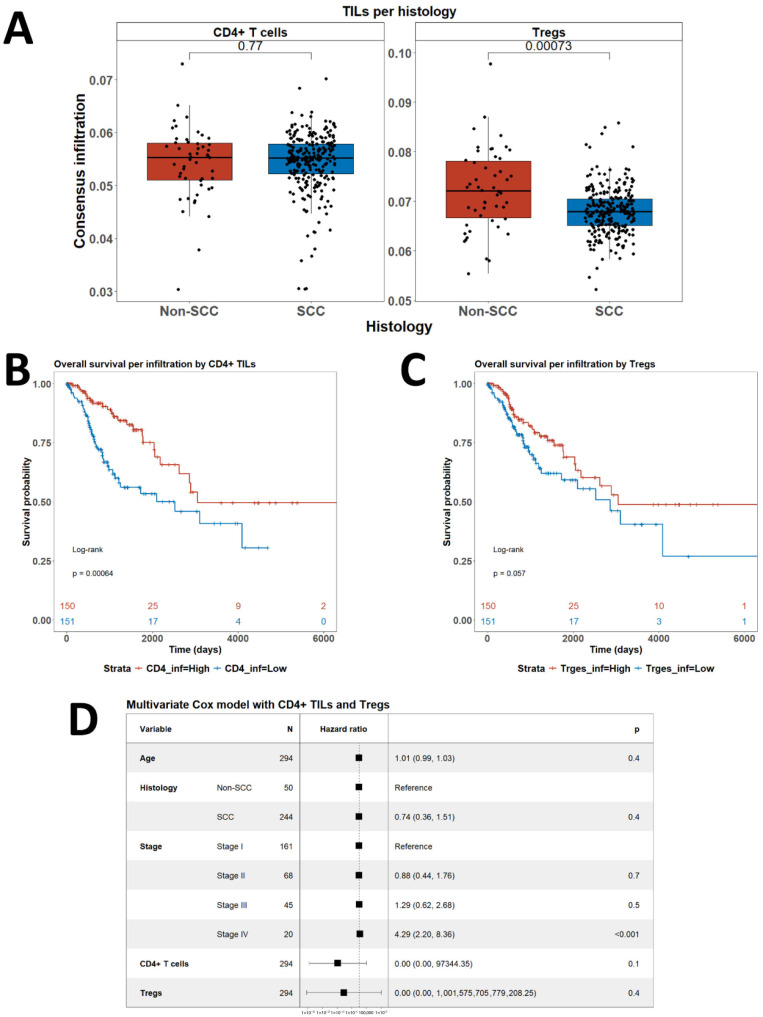
Analysis of consensus infiltration CD4+ T cells and Tregs data from TIMEDB. (**A**) Comparison of infiltration by histology. *p*-values are from two-sided *t*-test; (**B**) OS by level of infiltration with CD4+ T cells; (**C**) OS by level of infiltration with Tregs; (**D**) Forestplot of the outcomes of multivariate Cox model of OS analysis including CD4+ T cells and Tregs infiltration and available clinical covariables. In all statistical tests *p*-values below 0.05 were considered significant. Abbreviations: SCC—squamous cell carcinoma.

**Figure 6 biomedicines-13-02206-f006:**
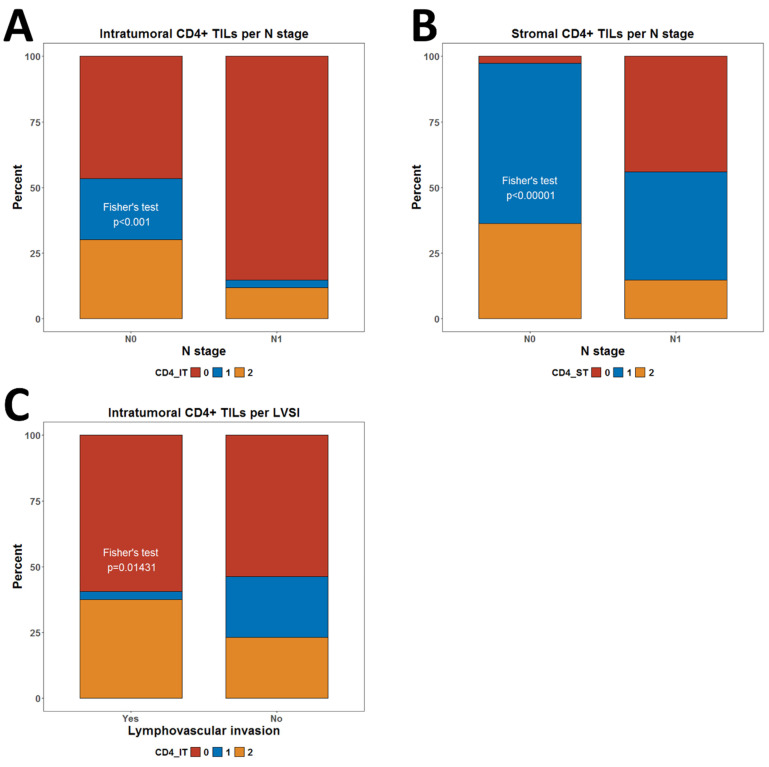
Comparison of levels of infiltration by CD4+ TILs by N status and LVSI. (**A**) Levels of CD4IT cells per N stage; (**B**) Levels of CD4ST cells per N stage; (**C**) Levels of CD4IT cells per LVSI status. All *p*-values are from Fisher’s exact test. *p*-values below 0.05 were considered significant. Abbreviations: LVSI—lymphovascular space invasion.

**Figure 7 biomedicines-13-02206-f007:**
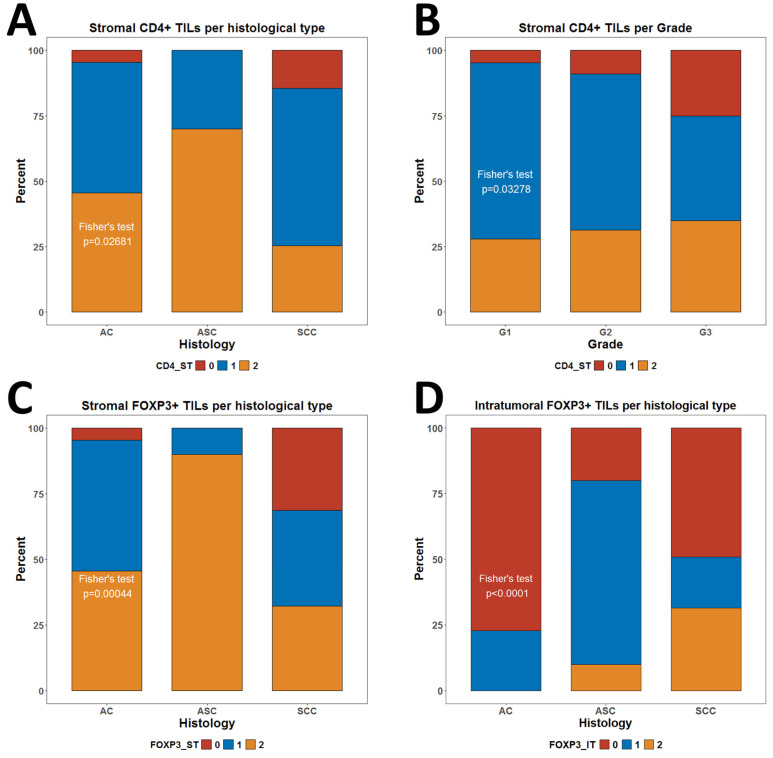
Comparison of levels of Infiltration by CD4+ TILs per histological subtype and grade of differentiation. (**A**) Levels of CD4ST cells per histological subtype; (**B**) Levels of CD4ST cells per grade of differentiation; (**C**) Levels of FOXP3ST cells per histological subtype; (**D**) Levels CD4IT cells per grade of differentiation. All *p*-values are from Fisher’s exact test. *p*-values below 0.05 were considered significant.

**Figure 8 biomedicines-13-02206-f008:**
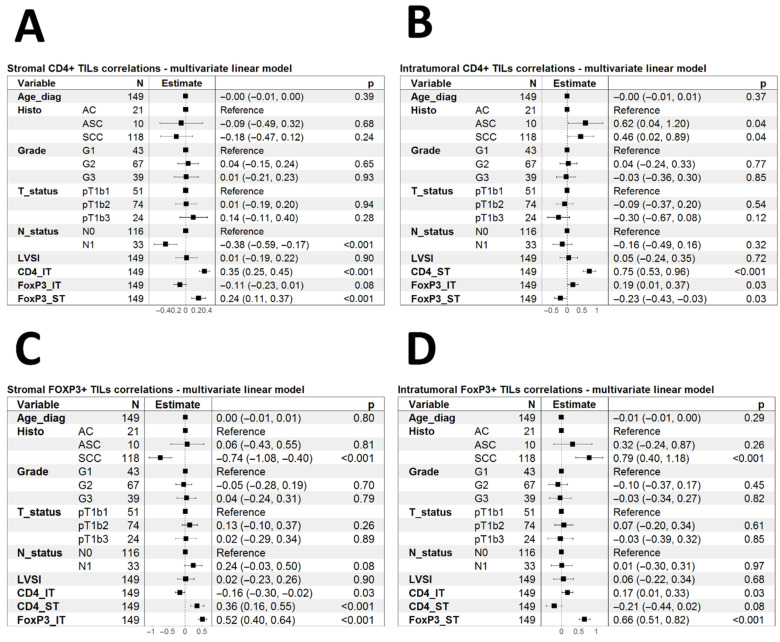
Forestplots summarizing the multivariate linear models of the association of any type of investigated infiltration with all other known covariables. (**A**) Model for CD4ST; (**B**) Model for CD4IT; (**C**) Model for FOXP3ST; (**D**) Model for FOXP3IT. *p*-values below 0.05 were considered significant.

**Figure 9 biomedicines-13-02206-f009:**
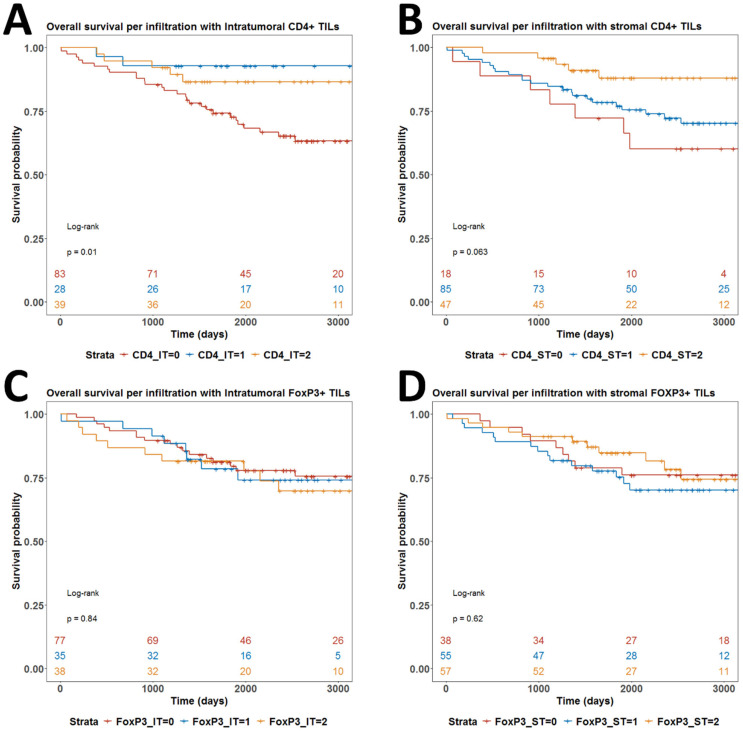
Univariate overall survival analysis. (**A**) Patients stratified per CD4IT infiltration; (**B**) Patients stratified per CD4ST infiltration; (**C**) Patients stratified per FOXP3IT infiltration; (**D**) Patients stratified per FOXP3ST infiltration. *p*-values are from log-rank test. *p*-values below 0.05 were considered significant.

**Figure 10 biomedicines-13-02206-f010:**
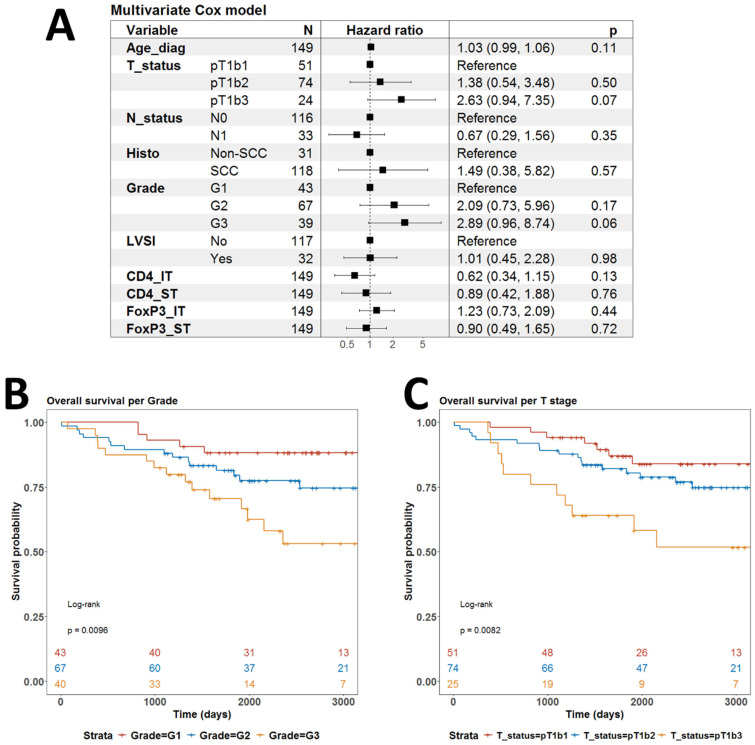
Multivariate and univariate overall survival analyses. (**A**) Forestplot of the outcome of a multivariate Cox model of overall survival with all covariables; (**B**) Univariate model of OS with patients stratified by grade of differentiation; (**C**) Univariate model of OS with patients stratified by T stage of differentiation. *p*-values in (**B**,**C**) are from log-rank test. *p*-values below 0.05 were considered significant.

**Figure 11 biomedicines-13-02206-f011:**
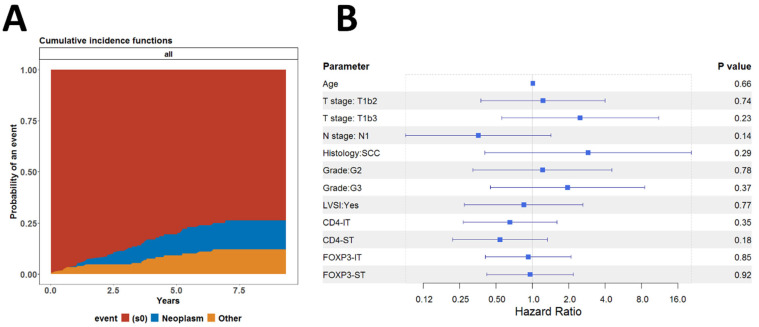
Regression analysis assuming competing risks model (**A**) Graphs of the estimated cumulative incidence functions for the risk of death due to either any malignant disease or any other cause; (**B**) Forestmodel summarizing the results of the Fine–Gray estimating the effect of all covariables on the cumulative incidence of death due to any malignancy.

**Table 1 biomedicines-13-02206-t001:** Summary of demographic and clinical characteristics of the patients included in the study. Abbreviations: G—grade; LVSI—lymphovascular space invasion; FIGO—International Federation of Gynecology and Obstetrics; AC—adenocarcinoma; ASC—adenosquamous carcinoma; SCC—squamous cell carcinoma.

Parameter	Count	Percent
Age		
≤50 years	78	52
>50 years	72	48
T stage		
T1b1	51	34
T1b2	74	49.33
T1b3	25	16.67
N stage		
N0	116	77.33
N1	34	22.67
FIGO stage		
FIGO I	116	77.33
FIGO III	34	22.67
Histology		
AC	22	14.67
ASC	10	6.66
SCC	118	78.67
Grade		
G1	45	30
G2	65	43.33
G3	40	26.67
LVSI		
Yes	32	21.33
No	117	78
Unknown	1	0.67
Total	150	100

**Table 2 biomedicines-13-02206-t002:** Distribution of CD4- and FOXP3-positive cells per subgroups of patients. *p*-values are from Fisher’s exact test. *p* < 0.05P were considered statistically significant and are denoted in red bold text. Abbreviations: FIGO—International Federation of Gynecology and Obstetrics; AC—adenocarcinoma; ASC—adenosquamous carcinoma; SCC—squamous cell carcinoma; G—grade; LVSI—lymphovascular space invasion; CD4IT—intratumoral T-helper cells; CD4ST—stromal T-helper cells FOXP3IT—intratumoral Forkhead box P3; FOXP3ST—stromal Forkhead box P3.

	CD4IT					CD4ST					FOXP3IT					FOXP3ST				
	0	1	2	Total	*p*-Value	0	1	2	Total	*p*-Value	0	1	2	Total	*p*-Value	0	1	2	Total	*p*-Value
Age																				
≤50 years	43	13	22	78	0.7167	7	43	28	78	0.324	33	24	21	78	** 0.03796 **	16	28	34	78	0.2433
>50 years	40	15	17	72		11	42	19	72		44	11	17	72		22	27	23	72	
Total	83	28	39	150		18	85	47	150		77	35	38	150		38	55	57	150	
T stage																				
T1b1	25	13	13	51	0.4952	4	28	19	51	0.6499	31	13	7	51	0.08425	13	20	18	51	0.6768
T1b2	41	12	21	74		11	41	22	74		31	18	25	74		16	27	31	74	
T1b3	17	3	5	25		3	16	6	25		15	4	6	25		9	8	8	25	
Total	83	28	39	150		18	85	47	150		77	35	38	150		38	55	57	150	
N stage																				
N0	54	27	35	116	** 0.00014 **	3	71	42	116	** 8.25 × 10^−9^ **	63	26	27	116	0.3464	32	38	46	116	0.1892
N1	29	1	4	34		15	14	5	34		14	9	11	34		6	17	11	34	
Total	83	28	39	150		18	85	47	150		77	35	38	150		38	55	57	150	
FIGO stage																				
FIGO I	54	27	35	116	** 0.00014 **	3	71	42	116	** 8.25 × 10^−9^ **	63	26	27	116	0.3464	32	38	46	116	0.1892
FIGO III	29	1	4	34		15	14	5	34		14	9	11	34		6	17	11	34	
Total	83	28	39	150		18	85	47	150		77	35	38	150		38	55	57	150	
Histology																				
AC	16	2	4	22	0.1775	1	11	10	22	** 0.02681 **	17	5	0	22	** 6.12 × 10^−5^ **	1	11	10	22	** 0.000441 **
ASC	3	2	5	10		0	3	7	10		2	7	1	10		0	1	9	10	
SCC	64	24	30	118		17	71	30	118		58	23	37	118		37	43	38	118	
Total	83	28	39	150		18	85	47	150		77	35	38	150		38	55	57	150	
Grade																				
G1	22	10	11	43	0.6464	2	29	12	43	** 0.03278 **	21	10	12	43	0.9259	9	19	15	43	0.7016
G2	35	13	19	67		6	40	21	67		37	15	15	67		20	22	25	67	
G3	26	5	9	40		10	16	14	40		19	10	11	40		9	14	17	40	
Total	83	28	39	150		18	85	47	150		77	35	38	150		38	55	57	150	
LVSI																				
Yes	19	1	12	32	** 0.01431 **	4	18	10	32	0.9589	13	9	10	32	0.3949	8	10	14	32	0.7415
No	63	27	27	117		13	67	37	117		63	26	28	117		29	45	43	117	
Total	82	28	39	149		17	85	47	149		76	35	38	149		37	55	57	149	

## Data Availability

The authors declare that all related data are available from the corresponding author upon reasonable request.

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
