# Peer review of "Clinical Significance of Tumor Infiltrating T-Helper and Regulatory Cells in Bulgarian Cervical Cancer Patients"

_biomedicines, 2025, doi:10.3390/biomedicines13092206_

Round 1

Reviewer 1 Report

Comments and Suggestions for Authors

Clinical significance of tumor infiltrating T helper and regulatory cells in Bulgarian cervical cancer patients

The purpose of this manuscript is to determine if lymphocyte infiltration can be used to gain insights into tumor progression.

While clearly representing a great deal of work, there are two major flaws in this study which require rejection of the manuscript at this stage.

1 - The authors separate out CD4+ cells (identified as T helper cells) and FOX3+ cells (identified as T regulatory cells) but T regulatory lymphocytes are a subset of CD4+ lymphocytes and can express both receptors.  This distinction is not clearly made or defended in the manuscript.

2 – Cervical cancer in humans (as the authors acknowledge) is most frequently associated with papilloma infections.  Without knowing the HPV status, the significance of the results is greatly diminished.

Beyond these limitations, the histologic analysis as illustrated by the images is inadequately explained. 

Who did the assessment?

How were random areas selected?

Why are there arrows pointing to background in the “absent” images in Figures 1 and 2?

Why does Figure 1A have a large (partially CD4+positive) lymphofollicular nodule in the adjacent stroma but still described as negative?

Author Response

Comment 1: The authors separate out CD4+ cells (identified as T helper cells) and FOX3+ cells (identified as T regulatory cells) but T regulatory lymphocytes are a subset of CD4+ lymphocytes and can express both receptors.  This distinction is not clearly made or defended in the manuscript.

Reply: We disagree that the distinction is not made in the manuscript. Separate staining for CD4 and FOXP3 was a standard approach in previous reports on CC patients, which were in depth discussed in the Discussion section. Please refer to lines 339-355 and 370-400.

Comment 2: Cervical cancer in humans (as the authors acknowledge) is most frequently associated with papilloma infections.  Without knowing the HPV status, the significance of the results is greatly diminished.

Reply: We disagree with this comment. Actually, the largest integrated molecular biology study in CC (TCGA) demonstrated that HPV-status did not correlate with defined molecular subtypes in squamous CC and therefore HPV status failed to be of prognostic relevance in multivariate models integrating molecular features. Please refer to: The Cancer Genome Atlas Research Network. Integrated genomic and molecular characterization of cervical cancer. Nature 543, 378–384 (2017). ttps://doi.org/10.1038/nature21386

Comment 3: Beyond these limitations, the histologic analysis as illustrated by the images is inadequately explained.

Reply: We edited all figures captions and added additional text as described below.

Comment 4: Who did the assessment?

Reply: We added the following text: The scoring of positive IHC stained cells for CD4 and FOXP3 was performed independently by two pathologists who had no information about the clinical and pathological data of the studied cases.

Comment 4: How were random areas selected?

Reply: We added the following text: Initially, the entire tumor area in each IHC stained slide was examined. Then CD4 + and FOXP3-positive lymphocytes were scored in the tumor and stroma of five randomly selected fields, at high magnification (high power fields - HPF), x400, without focusing in a hot-spot when there were positive cells.

Comment 5: Why are there arrows pointing to background in the “absent” images in Figures 1 and 2?

Reply: The arrows are pointing to TILs which were negative on CD4 staining for the sake of consistency between panels.

Comment 6: Why does Figure 1A have a large (partially CD4+positive) lymphofollicular nodule in the adjacent stroma but still described as negative?

Reply: The section demonstrates assessment of intraumoral and not of stromal TILs infiltration.

Reviewer 2 Report

Comments and Suggestions for Authors

Authors demonstrated that CD4IT or CD4ST are associated with biological and clinical features in cervical cancer with spatial analysis using conventional immunohistochemistry, inferring the pathological significance of CD4+T-cells infiltration into the tumor site in creation of immune-environment in cervical cancer. Although only the results of single staining of CD4 or FOXP3, their findings provide direction of the development surrogate biomarkers forecast on CD4+T-cell subsets for immunotherapy to cervical cancer.

Minor comments

  • FOXP3+ Treg infiltration was not associated with any clinical features. Please add the consideration to this reason.
  • Descriptions in "Discussion" are so long. It is recommended to move a part of description in "Discussion" to "Introduction".
  • Misspellings, Mistyping

Lines 100,113-116; FOXoxP3→FOXP3

Line 212; p values of “CD4IT (p= 0.0001) and CD4ST (p<0.0001)” are not consistent with those in Fig4A,B

Line 306; same the same→the same

Line 428; subgrupes→subgroups

Line 444; Different font is used in “modulation of the activity of CD4+ TILs”

Author Response

Comment 1: FOXP3+ Treg infiltration was not associated with any clinical features. Please add the consideration to this reason.

Reply: Actually, FOXP3 infiltration was dependent on histological subtype as shown in Figure 5 and mentioned in the discussion: Our work showed that this layer of pathobiological complexity is important as in our multivariate models stromal Tregs infiltration was significantly lower in SCC than in other histological subtypes. Contrary, intratumoral Tregs infiltration was significantly higher in SCC. In multivariate Cox models however neither stromal nor intratumoral Tregs had independent prognostic value. We added an additional consideration as requested: We did not observe other correlations of Tregs infiltration with any clinical features which might be due to the fact that the cohort was relatively uniform in terms of age and tumor size.

Comment 2: Descriptions in "Discussion" are so long. It is recommended to move a part of description in "Discussion" to "Introduction".

Reply: We shortened the discussion by deleting the in-depth description of common cancer ecotypes.

Comment 3: Lines 100,113-116; FOXoxP3→FOXP3

Reply: Corrected accordingly to FOXP3. Figure 5D was also corrected accordingly.

Comment 4: Line 212; p values of “CD4IT (p= 0.0001) and CD4ST (p<0.0001)” are not consistent with those in Fig4A,B

Reply: Corrected accordingly in the text to: “CD4IT (p<0.001) and CD4ST (p<0.00001)”

Comment 5: Line 306; same the same→the same

Reply: Corrected accordingly in the text to: “the same”

Comment 6: Line 428; subgrupes→subgroups

Reply: Corrected accordingly in the text to: “subgroups”

Comment 7: Line 444; Different font is used in “modulation of the activity of CD4+ TILs”

Reply: Font corrected accordingly.